# An additive Gaussian process regression model for interpretable non-parametric analysis of longitudinal data

Lu Cheng[1,2], Siddharth Ramchandran[1], Tommi Vatanen [3,4], Niina Lietzén[5], Riitta Lahesmaa[5], Aki Vehtari[1] & Harri Lähdesmäki[1]

Biomedical research typically involves longitudinal study designs where samples from individuals are measured repeatedly over time and the goal is to identify risk factors (covariates) that are associated with an outcome value. General linear mixed effect models are the standard workhorse for statistical analysis of longitudinal data. However, analysis of longitudinal data can be complicated for reasons such as difficulties in modelling correlated outcome values, functional (time-varying) covariates, nonlinear and non-stationary effects, and model inference. We present LonGP, an additive Gaussian process regression model that is specifically designed for statistical analysis of longitudinal data, which solves these commonly faced challenges. LonGP can model time-varying random effects and non-stationary signals, incorporate multiple kernel learning, and provide interpretable results for the effects of individual covariates and their interactions. We demonstrate LonGP's performance and accuracy by analysing various simulated and real longitudinal -omics datasets.

[1] Department of Computer Science, Aalto University School of Science, FI-00076 Aalto, Finland. [2] Microbiomes, Microbes and Informatics Group, Organisms and Environment Division, School of Biosciences, Cardiff University, Cardiff CF10 3AX, UK. [3] Broad Institute of MIT and Harvard, Cambridge, MA 02142, USA. [4] The Liggins Institute, University of Auckland, Auckland 1023, New Zealand. [5] Turku Centre for Biotechnology, University of Turku and Åbo Akademi University, FI-20520 Turku, Finland. Correspondence and requests for materials should be addressed to L.C. (email: lu.cheng.ac@gmail.com) or to H.L. (email: harri.lahdesmaki@aalto.fi)

Biomedical research often involves longitudinal studies where individuals are followed over a period of time and measurements are repeatedly collected from the subjects of the study. Longitudinal studies are effective in identifying various risk factors that are associated with an outcome, such as disease initiation, disease onset or any disease-associated molecular bio-marker. Characterisation of such risk factors is essential in understanding disease pathogenesis, as well as in assessing an individuals' disease risk, patient stratification, treatment choice evaluation, in a future personalised medicine paradigm, and planning disease prevention strategies.

There are several classes of longitudinal study designs, including prospective vs. retrospective studies and observational vs. experimental studies, and each of these can be implemented with a particular application-specific experimental design. Also, as the risk factors (or covariates) can be either static or time-varying, statistical analysis tools need to be versatile enough so that they can be appropriately tailored to every application. Traditionally, analysis of variance (ANOVA), general linear mixed effect models (LME), and generalised estimating equations are widely used in analysing longitudinal data due to their sim-plicity and interpretability[1]. Although numerous advanced extensions of these statistical techniques have been proposed, longitudinal data analysis is still complicated for several reasons, such as difficulties in choosing covariance structures to model correlated outcomes, handling irregular sampling times and missing values, accounting for time-varying covariates, choosing appropriate nonlinear effects, modelling non-stationary (ns) sig-nals, and accurate model inference.

Modern statistical methods for timeseries and longitudinal data analysis make less assumptions about the underlying data gen-erating mechanisms. These methods use predominantly non-parametric models, such as splines[2], and more recently latent stochastic processes, such as Gaussian processes (GP)[3,4]. While spline models can implement complex nonlinear functions, they are less efficient in modelling effects of covariate interactions. GP is a principled, probabilistic approach to learn non-parametric models, where nonlinearity is implemented through kernels[5]. A GP modelling framework is adopted in this work due to its flexibility and probabilistic formulation.

GPs have become a popular method for non-parametric modelling, especially for time-series data, and a wide variety of kernel functions have been proposed for different modelling tasks. A GP model can be made additive by defining the kernel function to be a sum of kernels. Similarly, a product of two or more kernels is also a valid kernel[5]. Thus, GPs can be made more interpretable and flexible by decomposing the kernel into a sum of individual and product (interaction) kernels much in the same way, conceptually, as with standard linear models. Here we can view the individual kernels as flexible nonlinear functions, which corresponds to the linear terms in linear regression. Plate[6] was among the first to formulate additive GPs by proposing a sum of univariate and multivariate kernels in an attempt to balance between model complexity and interpretability. Duvenaud et al.[7] considered an additive kernel that includes all input interaction terms and proposes a method for learning point estimates of kernel parameters by maximising the marginal likelihood. More complex kernel functions and structures were considered later[8]. Gilboa et al.[9] proposed Bayesian inference for additive GPs, whereas a hypothesis testing framework for nonlinear effects with GP was later proposed[10]. Bayesian semi-parametric models[4] and additive GP regression together with Bayesian inference meth-ods[11] were proposed in the context of longitudinal study designs. Schulam et al.[12] presented a method that combines linear com-ponents, spline components, and GP components to model a data set with a hierarchical structure. Computationally efficient model

inference for additive GP models (AGPM) using sparse approx-imations and variational inference was recently proposed[13].

We present LonGP, a flexible and interpretable non-parametric modelling framework together with a versatile software imple-mentation that solves commonly faced challenges in longitudinal data analysis. LonGP implements an additive GP regression model, with appropriate product kernels, that is specifically designed for longitudinal biomedical data with complex experi-mental designs. LonGP inherits the favourable features of GPs and multiple kernel learning. Our method extends previous GP (as well as linear mixed effect) models in several ways. Contrary to previous GP methods, LonGP implements a multi-level model that is conceptually similar to the commonly used linear models, and thus enables modelling individual-specific time-varying random effects, for example. LonGP also models ns signals using ns kernel functions and provides interpretable results for the effects of individual covariates and their interactions. We also develop a fully-Bayesian, predictive inference for LonGP and use that to carry out model selection, i.e. to identify covariates that are associated with a given study outcome value.

We demonstrate LonGP's performance and accuracy by ana-lysing various simulated and real longitudinal -omics data sets, including high-throughput longitudinal proteomics and metage-nomics data. We also compare LonGP with LME and GPs with automatic relevance determination (GP-ARD) kernel. LonGP with its full functionality is developed as an open-source software tool, which provides great convenience and flexibility of non-parametric longitudinal data analysis for applied research.

## Results

**Additive GP**. Linear models and their mixed effect variants have become a standard tool for longitudinal data analysis. However, a number of challenges still persist in longitudinal analysis, e.g. when data contains nonlinear and ns effects.

GP are a flexible class of models that have become popular in machine learning and statistics. Realizations from a GP correspond to random functions and, consequently, GPs naturally provide a prior for an unknown regression function that is to be estimated from data. Thus, GPs differ from standard regression models in that they define priors for entire nonlinear functions, instead of their parameters. While nonlinear effects can be incorporated into standard linear models by extending the basis functions e.g. with higher order polynomials, GPs can automatically detect any nonlinear as well as ns effects without the need of explicitly defining basis functions[5]. By definition, the prior probability density of GP function values $\boldsymbol{f}(X) = (f(\boldsymbol{x}_1),$ $f(\boldsymbol{x}_2), \cdots, f(\boldsymbol{x}_N))^T$ for any finite number of fixed input covariates $X = (\boldsymbol{x}_1, \boldsymbol{x}_2, ..., \boldsymbol{x}_N)$ (where $\mathbf{x}_i \in \mathcal{X}$) is defined to have a joint multivariate Gaussian distribution

$$\boldsymbol{f}(X) \sim N(\mathbf{0}, K_{X,X}(\boldsymbol{\theta})), \tag{1}$$

where elements of the $N$-by-$N$ covariance matrix are defined by the GP kernel function $[K_{X,X}(\boldsymbol{\theta})]_{i,j} = k(\boldsymbol{x}_i, \boldsymbol{x}_j|\boldsymbol{\theta})$ with parameters $\boldsymbol{\theta}$. Mean in Eq. (1) can in general depend on $X$ but zero mean is often assumed in practice. Covariance (also called the kernel function) of the normal distribution defines the smoothness of the function $f$, i.e. how fast the regression function can vary. Intuitively speaking, although GP is defined such that any finite-dimensional marginal has a Gaussian distribution, GP regression is a non-parametric method in the sense that the regression function $\boldsymbol{f}$ has no explicit parametric form. More formally, GP contains countably infinite many parameters that define the regression function, which are the function values $f$ at all possible inputs $\mathbf{x} \in \mathcal{X}$. For a comprehensive introduction to GPs we refer the reader to the book[5] and the Methods section.

GP models can be made more flexible and interpretable by making them additive, where the kernel (covariance) is a sum of kernels (covariances) and each kernel models the effect of individual covariates or their interactions, i.e. $f(\boldsymbol{x}) = f^{(1)}(\boldsymbol{x}) + \cdots + f^{(D)}(\boldsymbol{x})$. Intuitively one can think that each GP component $f^{(j)}$ now implements a nonlinear function that specifies the corresponding effect, and the overall effect of several covariates is then the sum of these nonlinear functions. This is achieved by using specific kernels for different types of covariates, such as squared exponential (se) kernel for continuous covariates, constant (co), binary (bi), and categorical (ca) kernels for discrete covariates, and products of these kernels for interaction terms. Moreover, ns signals can be accounted for by incorporating ns kernels.

Figure 1 shows an example where biomarker data $y$ is simulated from an AGPM that depends on continuous covariates age (age) and time from a disease event (diseaseAge) as well as discrete covariates ID (id) and location (loc) as follows: $y = f_{se}^{(1)}(\text{age}) + f_{ca \times se}^{(2)}(\text{id} \times \text{age}) + f_{ns}^{(3)}(\text{diseaseAge}) + f_{bi}^{(4)}(\text{loc}) + f_{bi \times se}^{(5)}(\text{loc} \times \text{age}) + f_{ca}^{(6)}(\text{id}) + \varepsilon$, where id identifies an individual and $\varepsilon$ is additive noise. In other words, the underlying regression function $f$ is decomposed into six separate (nonlinear) functions (Fig. 1, top row), and the measurements are corrupted by additive noise $\varepsilon$ (Fig. 1, top row, right panel). This example provides an intuitive illustration of nonlinear and ns effects of different kernels mentioned above. For example, continuous covariate age has a nonlinear effect on $y$ and similarly continuous covariate diseaseAge has a nonlinear and ns effect on $y$, where the largest change in the effect is localized at the time of disease onset. The overall cumulative effect is then defined by the sum of the individual nonlinear effects (Fig. 1, bottom row, second panel from right), and measurements of biomarker $y$ are corrupted by additive noise (Fig. 1, bottom row, right panel). In case a study contains other covariates or interaction terms, the additive GP regression provides a very flexible modelling framework that can be adjusted to a number of different applications.

Longitudinal studies typically involve two interrelated statistical questions: prediction of an outcome and model selection. While standard linear models are commonly constructed using hypothesis testing, here we develop a Bayesian predictive model selection method for the proposed AGPM that combines several state-of-the-art methodologies, including both Markov chain Monte Carlo (MCMC) sampling and approximate inference using central composite design (CCD). Furthermore, our model selection strategy involves assessing the predictive performance using cross-validation (with or without importance sampling), Bayesian bootstrap and a model search strategy for accurate model selection. For details of LonGP's statistical methodology, see Methods section. We tested LonGP on simulated data sets and two real data sets, including a longitudinal metagenomics[14] and a proteomics data sets[15], which are described below.

**Simulated data sets.** We first carried out a large simulation study to test and demonstrate LonGP's ability to correctly infer associations between covariates and target variables from longitudinal data. Here, we are primarily interested in answering two questions: is LonGP able to select the correct model as well as the correct covariates that were used to generate the data and can we detect disease-associated signals. We simulated nonlinear and ns -omics data sets from five different generative AGPM:

$$\text{AGPM1} : y = f_{ca}^{(1)}(\text{id}) + \varepsilon$$
$$\text{AGPM2} : y = f_{ca}^{(1)}(\text{id}) + f_{se}^{(2)}(\text{age}) + f_{ca \times se}^{(3)}(\text{id} \times \text{age}) + \varepsilon$$
$$\text{AGPM3} : y = f_{ca}^{(1)}(\text{id}) + f_{se}^{(2)}(\text{age}) + f_{ca \times se}^{(3)}(\text{id} \times \text{age}) + f_{bi}^{(4)}(\text{loc}) + f_{bi \times se}^{(5)}(\text{loc} \times \text{age}) + \varepsilon$$
$$\text{AGPM4} : y = f_{ca}^{(1)}(\text{id}) + f_{se}^{(2)}(\text{age}) + f_{ca \times se}^{(3)}(\text{id} \times \text{age}) + f_{ns}^{(4)}(\text{diseaseAge}) + \varepsilon$$
$$\text{AGPM5} : y = f_{ca}^{(1)}(\text{id}) + f_{se}^{(2)}(\text{age}) + f_{ca \times se}^{(3)}(\text{id} \times \text{age}) + f_{bi}^{(4)}(\text{loc}) + f_{bi \times se}^{(5)}(\text{loc} \times \text{age})$$
$$+ f_{ns}^{(6)}(\text{diseaseAge}) + \varepsilon$$

To set up our simulation scenario, we first use $P = 40$ individuals (which are divided into 20 cases and 20 controls for AGPM4 and AGPM5 due to the presence of sero effect in cases), each with $n_i = 13$ data points ranging from 0 to 36 months with an increment of three months, thus specifying the age covariate. Other covariates are randomly simulated using the following rules. The disease occurrence time is sampled uniformly from 0 to 36 months for each case subject and diseaseAge is computed accordingly. We make the effect of diseaseAge ns by transforming it with the sigmoid function from Eq. (16), such that majority of changes occur in the range of $-12$ to $+12$ months. The loc and gender are i.i.d. and sampled from a Bernoulli distribution with $p = 0.5$ for each individual, where gender and group act as irrelevant covariates. The continuous covariates are subjected to standardisation after being generated, such that the mean of each covariate is 0 and standard deviation is 1. We then sample latent function values and data from all the five models with the kernels described above (for details, see Methods), where the length-scales for continuous (standardised) covariates are set to 1 for the shared components and 0.8 for the interaction components.

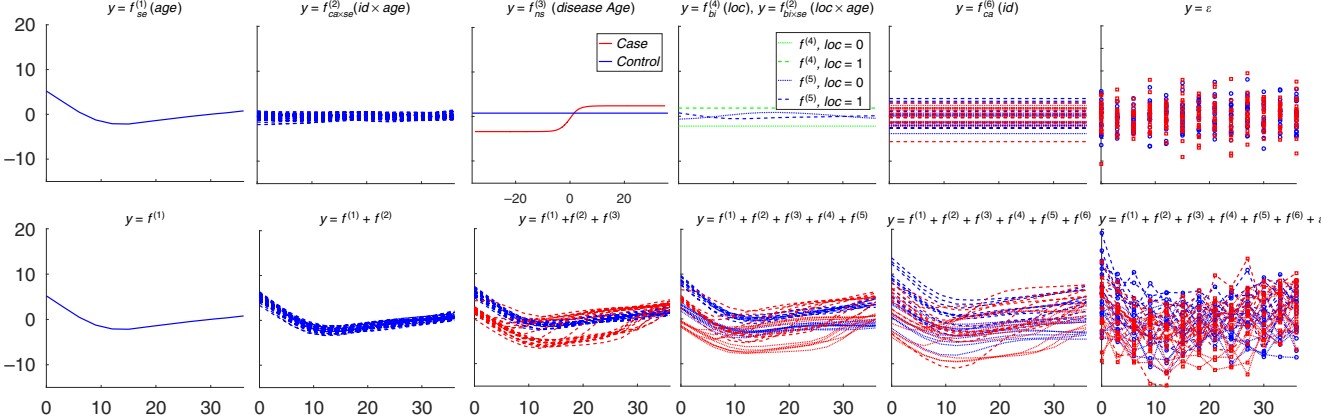

**Fig. 1** An additive Gaussian process (Simulated data). The *x*-axis is age by default except for the third figure in the top panel, which is the disease age. The top panel shows random functions drawn from different components, i.e. GPs of the specific kernels. The lower panel shows the cumulative effects of the different components. The bottom right panel shows the simulated data

We set the variances of each shared component to 4 and noise to 3, i.e. $\sigma^2_{\text{age}} = \sigma^2_{\text{diseaseAge}} = \sigma^2_{\text{loc}} = \sigma^2_{\text{id}} = 4$ and $\sigma^2_\varepsilon = 3$. With these specifications, we generate 100 data sets for each AGPM. A randomly generated longitudinal data set from AGPM5 is visualised in Fig. 1 (Note, the order of latent functions is changed for better visualisation).

In the inference, all covariates including irrelevant group and gender are used, which means that there are $2^5 = 32$ candidate models to choose from. Interaction terms are allowed for all covariates except for diseaseAge. Table 1 shows the distribution of selected models for each generating AGPM, with the numbers in bold font indicating correctly identified models. Table 1 shows that LonGP can achieve between 88% and 98% accuracy in inferring the correct model with these parameter settings. Results in Table 1 also show that it becomes more challenging to identify the correct model as the generating model becomes more complex, which is expected. LonGP can accurately detect the disease related signal as well, since the diseaseAge covariate is included in the final model for 97% of the simulation runs for both AGPM4 and AGPM5 models (see Table 1). Moreover, LonGP is notably specific in detecting the diseaseAge covariate as the percentage of false positives is only 0%, 1% and 0% for AGPM1, AGPM2 and AGPM3, respectively (see Table 1).

To better characterise LonGP's performance in different scenarios, we tested how the amount of additive noise affects the results. We varied the noise variance as $\sigma^2_\varepsilon \in \{1, 3, 5, 8\}$ and kept all other settings unchanged, effectively changing the signal to noise ratio or the effect size relative to the noise level. Figure 2a shows that the model selection accuracy increases consistently as the noise variance decreases. We next tested how the number of study subjects (i.e. the sample size $P$) affects the inference results. We set the number of case-control pairs to $\{(10, 10), (20, 20),$

$(30, 30), (40, 40)\}$ and kept all other settings unchanged. As expected, Fig. 2b shows how LonGP's model selection accuracy increases as the sample size increases. Similarly, LonGP maintains its high sensitivity and specificity in detecting the diseaseAge covariate across the additive noise variances and samples sizes considered here (see Supplementary Tables 1 and 2).

Finally, we also quantified how the sampling interval (i.e. the number of time points per individual) affects the inference results. We varied the sampling intervals as $\{2, 3, 4, 6\}$ (months) corresponding to $n_i \in \{19, 13, 10, 7\}$ time points for each individual and kept all other simulation settings unchanged. Supplementary Table 3 shows that, again, the model selection accuracy changes consistently with the number of measurement time points. Supplementary Table 4 shows that changing the sampling interval has a small but systematic effect on the sensitivity and specificity of detecting the diseaseAge covariate.

To demonstrate LonGP's performance relative to previous methods, we analysed the same simulated data sets using three traditional methods: (a) LME, (b) LME with second-order polynomial terms (LME-P), and (c) GP-ARD. We include GP-ARD in performance comparisons because it is the most commonly used method for assessing relevance of variables in GP regression. The ARD kernel contains an individual length-scale parameter for each input covariate, and the relevance of each covariate is determined by the estimated length-scale value, large (small) values indicate lower (higher) relevance. In LME and LME-P, the same effects in the generating models are considered as for LonGP. Specifically, individual variations are modelled as random effects and others are modelled as fixed effects. In GP-ARD, only shared effects are considered and interactions are not considered. See Supplementary Method 1 for detailed descriptions.

### Table 1 Model inference results

| Generating model | AGPM1 | AGPM2 | AGPM3 | AGPM4 | AGPM5 | Others | diseaseAge included | diseaseAge not included |
|---|---|---|---|---|---|---|---|---|
| AGPM1 | **98** | 2 | 0 | 0 | 0 | 0 | 0 | 100 |
| AGPM2 | 0 | **95** | 2 | 1 | 0 | 2 | 1 | 99 |
| AGPM3 | 0 | 0 | **95** | 0 | 0 | 5 | 0 | 100 |
| AGPM4 | 0 | 3 | 0 | **92** | 3 | 2 | 97 | 3 |
| AGPM5 | 0 | 0 | 3 | 8 | **88** | 1 | 97 | 3 |

The data is simulated with $P = 40$ individuals (20 cases and 20 controls), noise variance $\sigma^2_\varepsilon = 3$ and samples taken every 3 months. Rows show the number of times each model is inferred as the best model out of 100 Monte Carlo simulations for each generating model. 'Others' corresponds to all the other $32 - 5 = 27$ possible AGPM. The last two columns show the number of times if the diseaseAge covariate is included in the final model
*AGPM* Additive GP Models

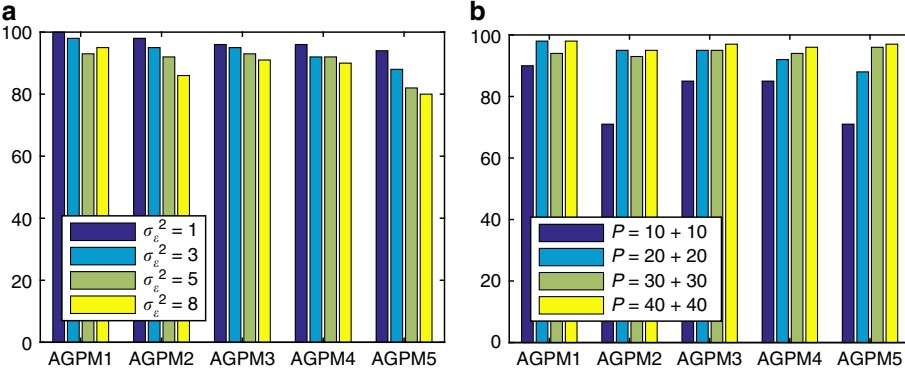

**Fig. 2** LonGP accuracy by varying noise and sample size. **a** Model selection accuracy as a function of noise variance. **b** Model selection accuracy as a function of sample size. AGPM stands for Additive GP Models. *y*-axis shows the number of times the correct model is inferred as the best model out of 100 Monte Carlo simulations (Simulated data sets)

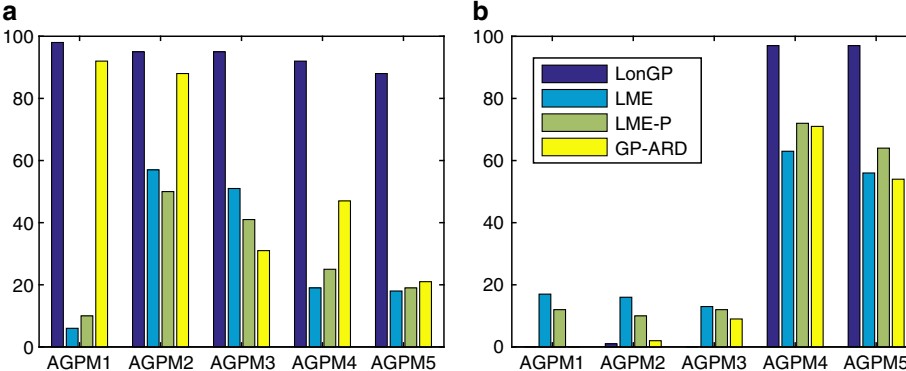

**Fig. 3** Methods comparison. **a** Model selection accuracy. **b** Disease effect detection accuracy. As in Table 1, y-axis shows **a** the number of times the correct model is inferred as the final model and **b** the number of times the diseaseAge covariate is included in the final model out of 100 Monte Carlo simulations (Simulated data sets). Note that disease effect is only expected for AGPM4 and AGPM5 in panel **b**. AGPM stands for Additive GP Models. LME-P stands for Linear Mixed Effect model with Polynomial terms. GP-ARD stands for GP with Automatic Relevance Determination kernel

Figure 3 shows the number of times the correct models were identified and the number of times the diseaseAge term was detected in the final model, for the same experiment settings as in Table 1. LonGP has a notably better accuracy than the traditional methods in selecting the correct model (Fig. 3a), as well as significantly better sensitivity (AGPM4-5 in Fig. 3b) and specificity (AGPM1-3 in Fig. 3b) in detecting the disease related effect. Full results of LME, LME-P, and GP-ARD over all simulated data sets are provided in Supplementary Note 1 and Supplementary Data 1.

Overall, our results suggest that LonGP can accurately infer the correct model structure and also detect a relatively weak disease related signal with as few as 10 case-control pairs and notable noise variance. Moreover, the model selection accuracy increases as the number of individuals (biological replicates), the number of time points, and signal to noise ratio increases.

**Longitudinal metagenomics data set**. We used LonGP to analyse a longitudinal metagenomics data set[14]. In this data set, 222 children from Estonia, Finland, and Russia were followed from birth until the age of three years through the collection of longitudinal stool samples, which were subsequently analysed by metagenomic sequencing. The aim of this study was to characterise the developing gut microbiome in infants from countries with different socio-economic status and to determine the key factors affecting the early gut microbiome development. Here, we model the microbial pathway profiles (i.e. total count of metagenomic reads mapping to bacterial genes involved in a pathway) quantifying the functional potential of the metagenomic communities. There are in total $N = 785$ metagenomic samples. To focus our analysis on pathways with sufficiently strong signal, we include in our analysis pathways that have been detected (i.e. at least one sequence read maps to genes of a pathway) in at least 64% (=500/785) of the samples. Let $c_{ij}$ denote the number of reads mapping to genes in the $j$th ($j = 1, \ldots, 394$) pathway in sample $i$ ($i = 1, \ldots, 785$) and $C_i$ is the total number of sequencing reads for sample $i$. The target variable is defined by $\log_2(c_{ij}/C_i \cdot \mathrm{median}(C_1, C_2, \ldots, C_N) + 1)$.

We selected the following 7 covariates for our additive GP regression based on their known interaction with the gut microbiome: age, bfo, caesarean, est, fin, rus, and id. bfo indicates whether an infant was breastfed at the time of sample collection; caesarean indicates if an infant was born by Caesarean section; est, fin, and rus are bi covariates indicating the home country of the study subjects (Estonia, Finland, and Russia, respectively); id

denotes the study subjects. We use SE kernel for age and bfo, ca kernel for id, and bi kernel for caesarean, est, fin, and rus. Interactions are allowed for all covariates except for bfo.

We applied LonGP to analyse each microbial pathway as a target variable separately and inferred the covariates for each target variable as described above. The selected models and explained variances of the components for all 394 pathways are available in Supplementary Data 2. A key discovery in Vatanen et al.[14] was that Lipid A biosynthesis pathway was significantly enriched in the gut microbiomes of Finnish and Estonian children compared to Russian children. Our analysis confirmed the linear model based analysis[14] by selecting the following model for Lipid A biosynthesis pathway: $y = f_{se}^{(1)}(\text{age}) + f_{se}^{(2)}(\text{bfo}) + f_{bi}^{(3)}(\text{rus}) + f_{ca}^{(4)}(\text{id}) + f_{bi \times se}^{(5)}(\text{rus} \times \text{age}) + f_{ca \times se}^{(6)}(\text{id} \times \text{age}) + \varepsilon$, which shows the difference between the Russian and Finnish study groups. Explained variance of bfo was 0.2% and bfo was thus excluded from the final model. Figure 4a shows the normalised Lipid A biosynthesis data together with the additive GP predictions using kernels $y = f_{se}^{(1)}(\text{age}) + f_{bi}^{(3)}(\text{rus}) + f_{bi \times se}^{(5)}(\text{rus} \times \text{age})$. The obtained model fit is similar to that reported by Vatanen et al.[14] with an exception that the apparent nonlinearity is captured by the AGPM, but otherwise the new model conveys the same information. Our analysis also identified many novel pathways with differences between Finnish, Estonian, and Russian microbiomes and is reported in Supplementary Data 2.

**Longitudinal proteomics data set**. We next analysed a longitudinal proteomics data set from a type 1 diabetes (T1D) study[15]. Liu et al. measured the intensities of more than 2000 proteins from plasma samples of 11 children who developed T1D and 10 healthy controls. For each child, nine longitudinal samples were analysed with the last sample for each case collected at the time of T1D diagnosis, resulting in a total of 189 samples. Detection of T1D associated autoantibodies in the blood is currently held as the best early marker that predicts the future development of T1D, and most of the individuals turning positive for multiple T1D autoantibodies will later on develop the clinical disease. Identifying early markers for T1D that would be detected even before the appearance of T1D associated autoantibodies is a grand challenge. It would allow early disease prediction and possibly even intervention.

Liu et al. used a linear mixed model with quadratic terms to detect proteins that behave differently between cases and controls. However, they only regressed on age since they did not take into

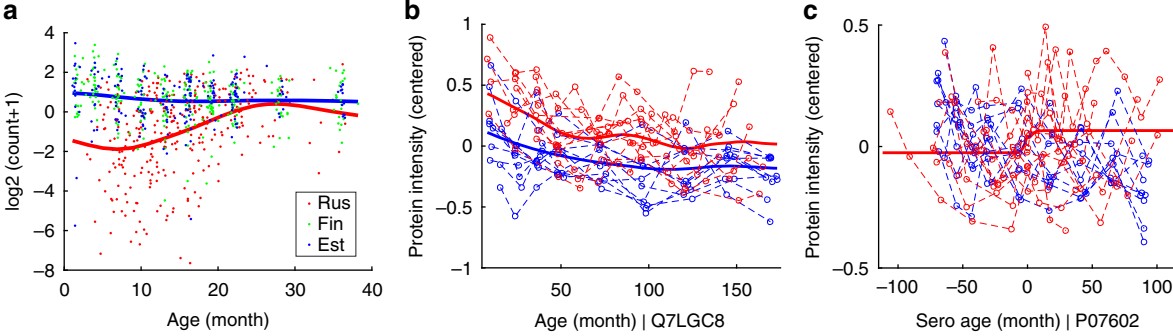

**Fig. 4** LonGP results. **a** LonGP regression results for Lipid A biosynthesis pathway (Metagenomics data set). Y-axis shows the $\log_2$ transformation of the normalised read counts of the samples. Russian, Finnish, and Estonian infant samples are depicted by the red, green, and blue colour dots, respectively. The blue line shows the nonlinear age trend of Finnish and Estonian infants. The red line shows the age trend of Russian infants. The red and blue lines are generated as the sum of components $y = f_{se}^{(1)}(\text{age}) + f_{bi}^{(3)}(\text{rus}) + f_{bi \times se}^{(5)}(\text{rus} \times \text{age})$. **b** Cumulative effect $y = f_{se}^{(1)}(\text{age}) + f_{bi}^{(2)}(\text{group}) + f_{bi \times se}^{(3)}(\text{group} \times \text{age})$ against real (centred) intensity of protein Q7LGC8 (Proteomics data set). Red lines are cases and blue lines are controls. **c** Predicted mean of the sero component for protein P07602 (Proteomics data set). The dashed red lines show the measurements of cases and the dashed blue lines are controls. x-axis indicates time from seroconversion and y-axis is the centred protein intensity. Mean seroconversion age of all cases (79.42 month) is used as the seroconversion age for controls. The solid red line corresponds to the mean of the seroconversion component $y = f_{ns}^{(4)}(\text{sero})$

account different seroconversion ages (age at the first detection of one or multiple T1D autoantibodies, as defined by Liu et al.[15]) of the cases and therefore could not model changes associated with seroconversion. We use LonGP to re-analyse this longitudinal proteomics data set[15] and try to find additional proteins with differing plasma expression profiles between cases and controls in general, as well as focusing on changes occurring close to seroconversion. Note that the age at which the T1D autoantibodies are detected is different for each individual. For each individual, the GP sero effect is then localized at the individual-specific seroconversion time point, making the sero effect consistent in the sero age coordinate but difficult (or impossible) to detect in the absolute age coordinate. The sero effect aims to detect nonlinear and ns effects that appear at specific times before and after the seroconversion, possibly near the time of the seroconversion. The modelling is done with the following covariates: age, sero (measurement time minus seroconversion time, see Methods), group (case or control), gender, and id. 1538 proteins with less than 50% missing values are kept for further analysis. We follow the same preprocessing steps[15] to get the normalised protein intensities. We use SE kernel for age, input warped ns SE kernel for sero, bi kernel for group as well as for gender, and ca kernel for id. Interactions are allowed for all covariates except for sero. The selected models and explained variances of each component for all 1538 proteins are reported in Supplementary Data 3.

We detected 38 proteins that are associated with the group covariate. In the original analyses by Liu et al.[15] [Table 1 and Supplementary Table S3], 18 of these proteins had the same temporal expression trend between the cases and the controls. As an example, we found the levels of Carbohydrate sulfotransferase 3 (UniProt Accession Q7LGC8) to be higher in cases than controls. The selected model for the protein is $y = f_{se}^{(1)}(\text{age}) + f_{bi}^{(2)}(\text{group}) + f_{bi \times se}^{(3)}(\text{group} \times \text{age}) + f_{ca \times se}^{(4)}(\text{id} \times \text{age}) + f_{ca}^{(5)}(\text{id}) + \varepsilon$. Figure 5 shows the contribution of each component and the cumulative effects. Figure 4b shows the cumulative effect $y = f_{se}^{(1)}(\text{age}) + f_{bi}^{(2)}(\text{group}) + f_{bi \times se}^{(3)}(\text{group} \times \text{age})$ against the real protein intensity to better visualise the predicted group difference.

We also detected altogether 47 proteins whose expression levels were changed relative to the time of seroconversion (sero covariate), with 20 of them having the same expression trend between the cases and the controls based on the analyses by Liu et al.[15]

(Table 1 and Supplementary Table S3). For two selected proteins, Prosaposin (Uniprot Accession P07602) and Opioid-binding protein/cell adhesion molecule (Uniprot Accession Q14982), protein expression levels were best explained by the LonGP model $y = f_{se}^{(1)}(\text{age}) + f_{ca \times se}^{(2)}(\text{id} \times \text{age}) + f_{ca}^{(3)}(\text{id}) + f_{ns}^{(4)}(\text{sero}) + \varepsilon$. Figure 4c shows the contribution of the *sero* component together with the real (centred) protein intensities as a function of seroconversion age for protein P07602. The sero component increases and then stabilises at a higher baseline after seroconversion in the cases. This is shown by the lower baseline of cases before seroconversion and higher baseline after seroconversion. Supplementary Fig. 1 shows the predicted mean of each component as well as the cumulative effects for protein P07602. Supplementary Fig. 2 shows a different type of sero effect for protein Q14982 where a temporary increase in protein intensity is observed close to the seroconversion time for many T1D patients, in contrast to the slowly decreasing age trend. Supplementary Fig. 3 shows the predicted individual components and the cumulative effects for protein Q14982.

## Discussion

General LME is a simple yet powerful modelling framework that has been widely accepted in biomedical literature. Still, applications of linear models can be challenging, especially when the underlying data generating mechanisms contain unknown non-linear effects and correlation structures or ns signals.

Here we have described LonGP, a non-parametric additive GP model for longitudinal data analysis, which we demonstrate to solve many of the commonly faced modelling challenges. As LonGP builds on GP regression, it can automatically handle irregular sampling time points and time-varying covariates. Missing values are also easily accounted for via bi mask kernels without any extra effort. More generally, LonGP provides a flexible framework to choose appropriate covariance structures for the correlated outcomes via the GP kernel functions, and the chosen kernels are properly adjusted to the given data by carrying out Bayesian inference for the kernel parameters. GP are known to be capable of approximating any continuous function. Thus, LonGP is applicable to any longitudinal data set. Furthermore, incorporating ns kernels into the kernel mixture easily adapts LonGP for ns signals. This allows us to model longitudinal phenomenon whose statistical properties are not time-shift invariant, which is especially useful for modelling e.g.

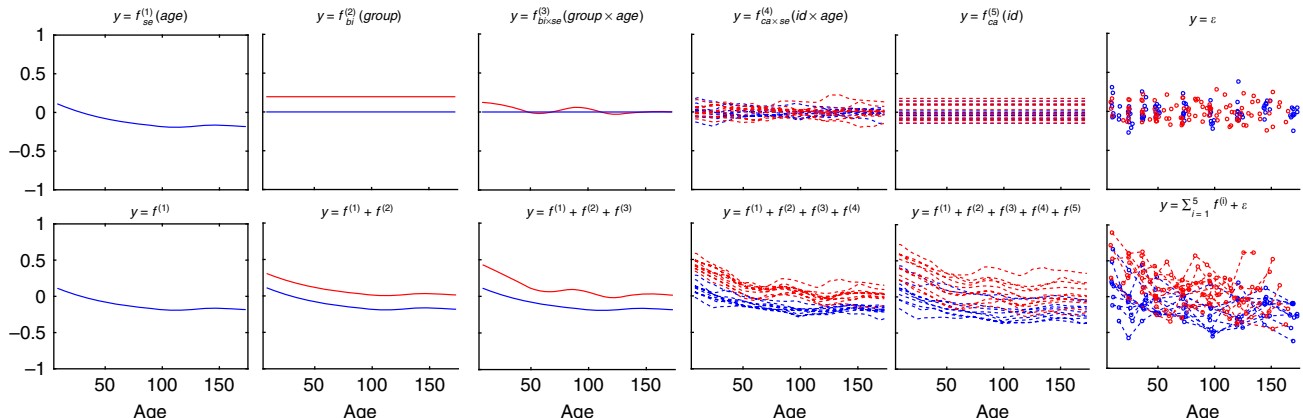

**Fig. 5** Predicted components and cumulative effect for protein Q7LGC8 (Proteomics data set). Top panel shows contributions of individual components and lower panel shows cumulative effects. Red lines are cases and blue lines are controls. Bottom right panel shows the (centered) data

pathophysiological mechanisms and changes that can have faster dynamics around a disease onset time than changes at other time points. While it is in principle possible to model ns signals with linear models, ns GP regression with Bayesian inference can be conveniently formulated and implemented using ns kernel functions, as we have shown here. Similar to standard GP regression methods, LonGP also provides predictions with quantified uncertainties (Eqs. (19) and (22)). As an example, corresponding to Fig. 4b, c, Supplementary Figs. 7 and 8 show the one standard deviation around the predictive mean. As the effect of individual covariates and their interactions can be quantified from the kernel mixture, LonGP provides an interpretable, non-parametric probabilistic analysis framework.

LonGP is equipped with an advanced Bayesian predictive inference method that utilises several recent, state-of-the-art techniques which make model inference accurate and improves running time especially for larger data sizes and more complex models. Finally, LonGP can be easily tailored for a variety of different longitudinal study designs. For example, multiple disease sub-types can be accounted for by using a ca kernel instead of a bi (case-control) kernel. Similarly, continuous phenotypes can be modelled using continuous kernels. For example, the ability to model the extensive, nonlinear age-associated changes observed in serum protein expression levels during early childhood[16] should improve the detection of disease-associated signals from such data.

LonGP has at least three features which makes it more efficient in our simulated data scenarios than the standard LME model. First, kernels automatically implement arbitrary nonlinear effects, whereas LME model is limited to linear (or second-order polynomial) effects. This is accentuated by having several nonlinear effects for individual covariates or their interactions. Moreover, characterising the posterior of the kernel parameters further improves LonGP's ability to identify nonlinear effects: instead of optimizing the kernel parameters to a given data set we also infer their uncertainty, and thus improve predicting new/unseen data points and inferring the covariate effects at the end. Second, LonGP contains ns effects that can be difficult to model using linear models. Third, LonGP naturally implements individual-specific time-varying random effects, which we consider relevant in modelling real biomedical longitudinal data sets, too.

Compared with traditional linear regression methods, LonGP is also useful in finding relatively weak signals that have an arbitrary shape. The dominant factor for Prosaposin (P07602) expression in the longitudinal proteomics data set[15] is age (explained variance 25%), while the disease related effect sero (explained variance 5.5%) is a minor factor, as shown in

Supplementary Fig. 1. Glucose-induced secretion of Prosaposin has been observed from a murine pancreatic beta-cell line[17]. Based on the LonGP analysis of the longitudinal proteomics data[15], the expression of Prosaposin in plasma decreases with age, but the baseline expression of the protein stabilises at a higher level in the cases after seroconversion. Similar changes were also detected by LonGP for secretogranin-3 (Q8WXD2), a protein with important functions in insulin secretory granules[18], and protein FAM3C (Q92520) also secreted from a murine beta-cell line in response to glucose[17]. However, no statistically significant differences were detected in the expression values of these proteins between cases and controls in the original analyses[15]. Seroconversion-associated changes in plasma levels of these proteins might reflect changes in the function or status of pancreatic beta-cells already before the onset of T1D. These, as well as other seroconversion-associated proteins revealed by our study provide a list of candidate proteins for further analysis with a more extensive sample size using, for example, targeted proteomics approaches. Similarly, in the longitudinal metagenomics data set[14], we also observed nonlinear effects for many of the covariates, some of which warrant further experimental studies. Revealing such disease related effects is essential in understanding mechanisms of disease progression and uncovering biomarkers for diagnostic purposes.

Apart from LME, only a few software packages exist for longitudinal data analysis. LonGP is accompanied by a software package that has all the functionality described here. Overall, supported by our results and open-source software implementation, we believe LonGP can be a valuable tool in longitudinal data analysis.

## Methods

**Notation**. We model target variables (gene/protein/bacteria/etc;) one at a time. Let us assume that there are $P$ individuals and there are $n_i$ time-series measurements from the $i$th individual. The total number of data points is thus $N = \sum_{i=1}^{P} n_i$. We denote the target variable by a column vector $\boldsymbol{y} = (y_1, y_2, \dots y_N)^T$ and the covariates by $X = (\boldsymbol{x}_1, \boldsymbol{x}_2, \dots, \boldsymbol{x}_N)$, where $\boldsymbol{x}_i = (x_{i1}, x_{i2}, \dots, x_{id})^T$ is a $d$-dimensional column vector and $d$ is the number of covariates. We denote the domain of the $j$th variable by $\mathcal{X}_j$ and the joint domain of all covariates is $\mathcal{X} = \mathcal{X}_1 \times \mathcal{X}_2 \times \dots \times \mathcal{X}_d$. In general, we use a bold font letter to denote a vector, an uppercase letter to denote a matrix, and a lowercase letter to denote a scale value.

**Gaussian process**. GP can be seen as a distribution of nonlinear functions[5]. For inputs $\boldsymbol{x}, \boldsymbol{x}' \in \mathcal{X}$, GP is defined as

$$f(\boldsymbol{x}) \sim GP(\mu(\boldsymbol{x}), k(\boldsymbol{x}, \boldsymbol{x}')), \qquad (2)$$

where $\mu(\boldsymbol{x})$ is the mean and $k(\boldsymbol{x}, \boldsymbol{x}')$ is a positive-semidefinite kernel function that

defines the covariance between any two realizations of $f(\boldsymbol{x})$ and $f(\boldsymbol{x}')$ by

$$k(\boldsymbol{x}, \boldsymbol{x}') = \text{cov}(f(\boldsymbol{x}), f(\boldsymbol{x}')), \tag{3}$$

which is called kernel for short. The mean is often assumed to be zero, i.e. $\mu(\boldsymbol{x}) \doteq 0$, and the kernel has parameters $\boldsymbol{\theta}$, i.e. $k(\boldsymbol{x}, \boldsymbol{x}'|\boldsymbol{\theta})$. For any finite collection of inputs $X = (\boldsymbol{x}_1, \boldsymbol{x}_2, ..., \boldsymbol{x}_N)$, the function values $\boldsymbol{f}(X) = (f(\boldsymbol{x}_1), f(\boldsymbol{x}_2), ..., f(\boldsymbol{x}_N))^T$ have joint multivariate Gaussian distribution

$$\boldsymbol{f}(X) \sim N(\boldsymbol{0}, K_{X,X}(\boldsymbol{\theta})), \tag{4}$$

where elements of the $N$-by-$N$ covariance matrix are defined by the kernel $[K_{X,X}(\boldsymbol{\theta})]_{i,j} = k(\boldsymbol{x}_i, \boldsymbol{x}_j|\boldsymbol{\theta})$.

We use the following hierarchical GP model

$$\begin{aligned}
\boldsymbol{\theta} &\sim \pi(\boldsymbol{\phi}) \\
\boldsymbol{f} &\sim N(\boldsymbol{0}, K_{X,X}(\boldsymbol{\theta})) \\
\boldsymbol{y} &\sim N(\boldsymbol{f}, \sigma_\varepsilon^2 I),
\end{aligned} \tag{5}$$

where $\pi(\boldsymbol{\phi})$ defines a prior for the kernel parameters (including $\sigma_\varepsilon^2$), $\sigma_\varepsilon^2$ is the noise variance, and $I$ is the $N$-by-$N$ identity matrix. For a Gaussian noise model, we can marginalise $\boldsymbol{f}$ analytically[5]

$$\begin{aligned}
p(\boldsymbol{y}|X, \boldsymbol{\theta}) &= \int p(\boldsymbol{y}|\boldsymbol{f}, X, \boldsymbol{\theta}) p(\boldsymbol{f}|X, \boldsymbol{\theta}) d\boldsymbol{f} \\
&= N(\boldsymbol{0}, K_{X,X}(\boldsymbol{\theta}) + \sigma_\varepsilon^2 I).
\end{aligned} \tag{6}$$

**Additive GP.** To define a flexible and interpretable model, we use the following AGPM with $D$ kernels

$$\begin{aligned}
f(\boldsymbol{x}) &= f^{(1)}(\boldsymbol{x}) + f^{(2)}(\boldsymbol{x}) + \cdots + f^{(D)}(\boldsymbol{x}) \\
y &= f(\boldsymbol{x}) + \varepsilon,
\end{aligned} \tag{7}$$

where each $f^{(j)}(\boldsymbol{x}) \sim \text{GP}(0, k^{(j)}(\boldsymbol{x}, \boldsymbol{x}'|\boldsymbol{\theta}^{(j)}))$ is a separate GP with kernel-specific parameters $\boldsymbol{\theta}^{(j)}$ and $\varepsilon$ is the additive Gaussian noise. By definition, for any finite collection of inputs $X = (\boldsymbol{x}_1, \boldsymbol{x}_2, ..., \boldsymbol{x}_N)$, each GP $\boldsymbol{f}^{(j)}(X)$ follows a multivariate Gaussian distribution. Since a sum of multivariate Gaussian random variables is still Gaussian, the latent function $\boldsymbol{f}$ also follows a multivariate Gaussian distribution. Denote $\boldsymbol{\Theta} = (\boldsymbol{\theta}^{(1)}, \boldsymbol{\theta}^{(2)}, ..., \boldsymbol{\theta}^{(D)}, \sigma_\varepsilon^2)$, then the marginal likelihood for the target variable $\boldsymbol{y}$ is

$$p(\boldsymbol{y}|X, \boldsymbol{\Theta}) = N\left(\boldsymbol{0}, \sum_{j=1}^{D} K_{X,X}^{(j)}(\boldsymbol{\theta}^{(j)}) + \sigma_\varepsilon^2 I\right), \tag{8}$$

where the latent function $\boldsymbol{f}$ has been marginalised out as in Eq. (6). To simplify notation, we define

$$K_{\boldsymbol{y}}(\boldsymbol{\Theta}) = \sum_{j=1}^{D} K_{X,X}^{(j)}(\boldsymbol{\theta}^{(j)}) + \sigma_\varepsilon^2 I. \tag{9}$$

For the purposes of identifying covariate subsets that are associated with a target variable, we assume that each GP depends only on a small subset of covariates $f^{(j)}(\boldsymbol{x}) : \mathcal{X}^{(j)} \to \mathcal{X}$, where $\mathcal{X}^{(j)} = \prod \mathcal{X}_i, i \in I_j \subseteq \{1, ..., d\}$ and $\mathcal{Y}$ is the domain for target variable. $I_j$ are indices of the covariates associated with the $j$th kernel.

**Kernel functions for covariates.** Longitudinal biomedical studies typically include a variety of continuous, ca, and bi covariates. Typical continuous covariates include age, time from a disease event (sampling time point minus disease event time point), and season (time from beginning of a year). Typical ca or bi covariates include group (case or control), gender, and id (id of an individual). In practice, to set up the AGPM, one needs to choose appropriate kernels for different covariates and their subsets (or interactions). We designed the following kernels to reflect common domain knowledge of longitudinal study designs, which covers most common modelling needs. Note that all kernels, including interactions, are automatically determined given a set of input covariates according to the algorithm in Supplementary Method 2.

**Stationary kernels.** In LonGP, we use the following specific stationary kernels which only involve one or two covariates. se kernel for continuous covariates

$$k_{\text{se}}(x_i, x_j|\boldsymbol{\theta}_{\text{se}}) = \sigma_{\text{se}}^2 \exp\left(-\frac{(x_i - x_j)^2}{2\ell_{\text{se}}^2}\right), \tag{10}$$

where $\ell_{\text{se}}$ is the length-scale parameter, $\sigma_{\text{se}}^2$ is the magnitude parameter and $\boldsymbol{\theta}_{\text{se}} = (\ell_{\text{se}}, \sigma_{\text{se}}^2)$. Length-scale $\ell_{\text{se}}$ controls the smoothness and magnitude parameter $\sigma_{\text{se}}^2$ controls the magnitude of the kernel.

Periodic kernel for continuous covariates

$$k_{\text{pe}}(x_i, x_j|\boldsymbol{\theta}_{\text{pe}}) = \sigma_{\text{pe}}^2 \exp\left(-\frac{2\sin^2(\pi(x_i - x_j)/\gamma)}{\ell_{\text{pe}}^2}\right), \tag{11}$$

where $\ell_{\text{pe}}$ is the length-scale parameter, $\sigma_{\text{pe}}^2$ is the magnitude parameter, $\gamma$ is the

period parameter, and $\boldsymbol{\theta}_{\text{pe}} = (\ell_{\text{pe}}, \sigma_{\text{pe}}^2, \gamma)$. Length-scale $\ell_{\text{pe}}$ controls the smoothness, $\sigma_{\text{pe}}^2$ controls the magnitude, and $\gamma$ is the period of the kernel. In our model, $\gamma$ corresponds to a year.

co kernel

$$k_{\text{co}}(x_i, x_j|\boldsymbol{\theta}) = \sigma_{\text{co}}^2, \tag{12}$$

where $\boldsymbol{\theta} = (\sigma_{\text{co}}^2)$ is the magnitude parameter of the co signal.

ca kernel for discrete-valued covariates

$$k_{\text{ca}}(x_i, x_j) = \begin{cases} 1, & \text{if } x_i = x_j \\ 0, & \text{otherwise.} \end{cases} \tag{13}$$

bi (mask) kernel for bi covariates

$$k_{\text{bi}}(x_i, x_j) = \begin{cases} 1, & \text{if } x_i = 1 \text{ and } x_j = 1 \\ 0, & \text{otherwise.} \end{cases} \tag{14}$$

Product kernel between any two valid kernels, such as $k_{\text{bi}}(\cdot)$ and $k_{\text{se}}(\cdot)$ (similarly for any other pair of kernels)

$$k_{\text{bi} \times \text{se}}(\cdot) = k_{\text{bi}}(x_{ip}, x_{jp}|\boldsymbol{\theta}_{\text{bi}}^{(p')}) k_{\text{se}}(x_{iq}, x_{jq}|\boldsymbol{\theta}_{\text{se}}^{(q')}), \tag{15}$$

where $\boldsymbol{\theta}_{\text{bi}}^{(p')}$ and $\boldsymbol{\theta}_{\text{se}}^{(q')}$ are kernel parameters for the $p$th and $q$th covariates, respectively.

**Non-stationary (ns) kernel.** It may be realistic to assume that the target variable (e.g. a protein) changes rapidly only near a special event, such as disease initiation or onset. This poses a challenge for GP modelling with se kernel since the kernel is stationary: changes are homogeneous across the whole time window. ns GPs can be implemented by using special ns kernels, such as the neural network kernel, by defining the kernel parameters to depend on input covariates[19–21] or via input or output warpings[22]. We propose to use the input warping approach and define a bijective mapping $\omega:(-\infty, +\infty) \to (-c, c)$ for a continuous time/age covariate $t$ as

$$\omega(t) = 2c \cdot \left(-0.5 + \frac{1}{1 + e^{-a(t-b)}}\right), \tag{16}$$

where $a$, $b$, and $c$ are predefined parameters: $a$ controls the size of the effective time window, $b$ controls its loc, and $c$ controls the maximum range. The ns kernel is then defined as

$$k_{\text{ns}}(t, t'|\boldsymbol{\theta}_{\text{se}}) = \sigma_{\text{se}}^2 \exp\left(-\frac{(\omega(t) - \omega(t'))^2}{2\ell_{\text{se}}^2}\right), \tag{17}$$

where $\boldsymbol{\theta}_{\text{se}}$ are the parameters of the SE kernel.

Supplementary Fig. 4 shows an example transformation with $a = 0.5$, $b = 0$ and $c = 40$, where we limit the disease related change to be within one year of the disease event. Effectively, all changes in the transformed space correspond approximately to $\pm 12$ month time window in the original space. Supplementary Fig. 5 shows randomly sampled functions using stationary and ns SE kernels with the same kernel parameters. The ns SE kernel naturally models signals that are spike-like or exhibit a level difference between before and after the disease event, which can be interpreted as a permanent disease effect.

The same parameters as Supplementary Fig. 4 are used for ns kernels in all experiments of the Results section.

**Kernel specification in practice.** The data sets analysed in this work include 11 covariates and covariate pairs which we model using the following kernels (see next section for prior specifications). age: The shared age effect is modelled with a slowly changing stationary SE kernel. Time from a disease event or diseaseAge: We use the product of the bi kernel and the ns SE kernel (assuming cases are coded as 1 and controls as 0). season: We assume that the target variable exhibits an annual period and is modelled with the periodic kernel. group: We model a baseline difference between the cases and controls, which corresponds to average difference between the two groups, using the product of the bi kernel and the co kernel. gender: We use the same kernel as for group covariate. loc: bi covariate indicating if an individual comes from a certain loc. We use the same kernel as for group covariate. id: We assume baseline differences between different individuals and model that by the product of the categorical kernel and the co kernel. group × age: We assume that the differences between cases and controls varies across age. That difference is modelled by the product of the bi kernel and the stationary SE kernel. gender × age: The same kernel as for group × age is used for this interaction term. It implements a different age trend for males and females. id × age: We assume different individuals exhibit different age trends. This longitudinal random effect is modelled by the product of the ca kernel and the SE kernel. This kernel is especially helpful for modelling individuals with outlying data points. group × gender: This interaction term assumes that male (or female) cases have a baseline difference compared to others. The product of two bi kernels and the co kernel is used. Although discrete covariates are modelled as a product of the co kernel and the bi or ca kernel, the co kernel is not explicitly included in our notation.

In practice, we often observe missing values in the covariates. Missing values can be due to technical problems in measurements or because some covariates may not be applicable for certain samples, e.g. diseaseAge is not applicable to controls

since they do not have a disease. In LonGP, we construct a bi flag vector for each covariate. The missing values are flagged as 0 and non-missing values are flagged as 1. Then, we construct a bi kernel for this flag vector and multiply it with any kernel that involves the covariate. Consequently, any kernel involving a missing value is evaluated to 0, which means that their contribution to the target variable is 0. All missing values are handled in this way by default and we do not use any extra notations for it. Interaction terms always refer to product kernels with non-missing values, assuming missing values are already handled.

**Prior specifications**. Before the actual GP regression, we standardise the target variable and all continuous covariates such that the mean is zero and the standard deviation is one. This helps in defining generally applicable priors for the kernel parameters. After the GP regression, the predictions are transformed back to the original scale. We visualise the results in the original scale after centering the data by subtracting the mean.

We define a prior $p(\Theta) = \prod_{j=1}^{D} p(\boldsymbol{\theta}^{(j)}) \times p(\sigma_{\epsilon}^2)$ for the kernel parameters as follows. For continuous covariates without interactions, we use the log normal prior ($\mu = 0$ and $\sigma^2 = (\log(1) - \log(0.1))^2/4$) for the length-scales ($\ell_{se}$ and $\ell_{pe}$) and the square root student-$t$ prior ($\mu = 0$, $\sigma^2 = 1$, and $\nu = 20$) for the magnitude parameters ($\sigma_{se}^2$ and $\sigma_{pe}^2$). This length-scale prior penalises small length-scales such that smoothness less than 0.1 has very small probability and the mode is approximately at 0.3. For continuous covariates with interactions, the prior for the magnitude parameters is the same as for without interactions and the half truncated student-$t$ prior ($\mu = 0$, $\sigma^2 = 1$, $\nu = 4$) is used for the length-scale, which allows smaller length-scales.

Scaled inverse chi-squared prior ($\sigma^2 = 0.01$ and $\nu = 1$) is used for the noise variance parameter $\sigma_{\epsilon}^2$. The period parameter $\gamma$ of the periodic kernel is predefined by the user. Square root student-$t$ prior ($\mu = 0$, $\sigma^2 = 1$, and $\nu = 4$) is used for the magnitude parameter $\sigma_{co}^2$ of all co kernels. Supplementary Fig. 6 visualises all the above-described priors with their default hyperparameter values.

**Model inference and prediction**. Given the AGPM, we are next interested in the posterior inference of the model conditioned on data ($\boldsymbol{y}$, $X$). Assume, for now, that for each additive component $f^{(j)}$ the kernel $k^{(j)}(\cdot)$, its inputs $\mathcal{X}^{(j)}$ and prior are specified. We use two different inference methods, MCMC and a deterministic evaluation of the posterior with the CCD.

For MCMC we use the slice sampler as implemented in the GPStuff package[23,24] to sample the parameter posterior

$$p(\Theta|\boldsymbol{y}, X) \propto p(\boldsymbol{y}|X, \Theta)p(\Theta), \tag{18}$$

where the likelihood is defined in Eq. (8). After convergence checking from 4 independent Markov chains (details in Supplementary Method 3), we obtain $S$ posterior samples $\{\Theta_s\}_{s=1}^S$, where $\Theta_s = (\boldsymbol{\theta}_s^{(1)}, \boldsymbol{\theta}_s^{(2)}, ..., \boldsymbol{\theta}_s^{(D)}, \sigma_{\epsilon,s}^2)$. We use the posterior samples to approximate the predictive density for test data $X^* = (\boldsymbol{x}_1^*, \boldsymbol{x}_2^*, ..., \boldsymbol{x}_n^*)$

$$\begin{aligned} p(\boldsymbol{f}^*|\boldsymbol{y}, X, X^*) &= \int p(\boldsymbol{f}^*|\boldsymbol{y}, X, X^*, \Theta)p(\Theta|\boldsymbol{y}, X)d\Theta \\ &\approx \frac{1}{S}\sum_{s=1}^{S} p(\boldsymbol{f}^*|\boldsymbol{y}, X, X^*, \Theta_s) \\ &= \frac{1}{S}\sum_{s=1}^{S} N(\boldsymbol{\mu}_s, \Sigma_s), \end{aligned} \tag{19}$$

where

$$\boldsymbol{\mu}_s = K_{X^*,X}(\Theta_s)K_{\boldsymbol{y}}(\Theta_s)^{-1}\boldsymbol{y} \tag{20}$$

$$\Sigma_s = K_{X^*,X^*}(\Theta_s) - K_{X^*,X}(\Theta_s)K_{\boldsymbol{y}}(\Theta_s)^{-1}K_{X,X^*}(\Theta_s) \tag{21}$$

are the standard GP prediction equations adapted to additive GPs with $K_{X^*,X}(\Theta_s) = \sum_{j=1}^{D} K_{X^*,X}^{(j)}(\boldsymbol{\theta}_s^{(j)})$ encoding the sum of cross-covariances between the inputs $X$ and test data points $X^*$ ($K_{X^*,X^*}$ is defined similarly) and $K_{\boldsymbol{y}}(\Theta_s)$ is defined in Eq. (9).

As an alternative approach to slice sampling for higher dimensional models, we also use a deterministic finite sum using the CCD to approximate the predictive densities for GPs[25,26]. CCD assumes a split-Gaussian posterior $q(\cdot)$ for (log-transformed) parameters $\boldsymbol{\gamma} = \log(\Theta)$ and defines a set of $R$ points $\{\boldsymbol{\gamma}_r\}_{r=1}^R$ (fractional factorial design, the mode, and so-called star points along whitened axes) to estimate the predictive density with a finite sum

$$\begin{aligned} p(\boldsymbol{f}^*|\boldsymbol{y}, X, X^*) &\approx \sum_{r=1}^{R} p(\boldsymbol{f}^*|\boldsymbol{y}, X, X^*, \boldsymbol{\gamma}_r)q(\boldsymbol{\gamma}_r)\Delta_r \\ &= \sum_{r=1}^{R} N(\boldsymbol{\mu}_r, \Sigma_r)q(\boldsymbol{\gamma}_r)\Delta_r, \end{aligned} \tag{22}$$

where $N(\boldsymbol{\mu}_r, \Sigma_r)$ is computed as in Eqs. (20), (21), $q(\boldsymbol{\gamma}_r)$ is the split-Gaussian posterior, and $\Delta_r$ are the area weights for the finite sum[26].

Predictions and visualisations for an individual kernel $k^{(j)}$ ($1 \le j \le D$) are obtained by replacing $\boldsymbol{\mu}_s$ and $\Sigma_s$ in Eqs. (19) and (22) with

$$\boldsymbol{\mu}_s^{(j)} = K_{X^*,X}^{(j)}(\boldsymbol{\theta}_s^{(j)})K_{\boldsymbol{y}}(\Theta_s)^{-1}\boldsymbol{y} \tag{23}$$

$$\Sigma_s^{(j)} = K_{X^*,X^*}^{(j)}(\boldsymbol{\theta}_s^{(j)}) - K_{X^*,X}^{(j)}(\boldsymbol{\theta}_s^{(j)})K_{\boldsymbol{y}}(\Theta_s)^{-1}K_{X,X^*}^{(j)}(\boldsymbol{\theta}_s^{(j)}). \tag{24}$$

Similarly, predictions for a subset of kernels are obtained by replacing $K_{X^*,X}^{(j)}(\boldsymbol{\theta}_s^{(j)})$ and $K_{X^*,X^*}^{(j)}(\boldsymbol{\theta}_s^{(j)})$ with the relevant sums.

**Model comparison**. We have described how to build and infer an AGPM for a given target variable using a set of kernels and a set of covariates for each kernel. A model $M$ can be specified by a 3-tuple $(D, \{k^{(j)}\}_{j=1}^D, \{I_j\}_{j=1}^D)$, where $D \ge 1$. However, all covariates may not be relevant for the prediction task and often the scientific question is to identify a subset of the covariates that are associated with the target variable. For model selection, we use two cross-validation variants and Bayesian bootstrap as described below.

**Leave-one-out cross-validation**. We use leave-one-out cross-validation (LOOCV) to compare the models when a continuous covariate such as age, diseaseAge or season is added to a model. In this case, a single time point of an individual is left out as test data and the rest are kept as training data. We use MCMC to infer the parameters of a given model and calculate the following leave-one-out predictive density

$$p(y_i|\boldsymbol{y}_{-i}, X, M) = \int p(y_i|\Theta, X, M)p(\Theta|\boldsymbol{y}_{-i}, X, M)d\Theta, \tag{25}$$

where $\boldsymbol{y}_{-i} = \boldsymbol{y}|y_i$ and $\Theta$ are the parameters of the GP model $M$. This can be calculated by setting $\boldsymbol{f}^* \leftarrow y_i$, $X^* \leftarrow \boldsymbol{x}_i$, $\boldsymbol{y} \leftarrow \boldsymbol{y}_{-i}$ and $X \leftarrow X|\boldsymbol{x}_i$ in Eq. (19). The standard LOOCV would require us to run the inference $N$ times, which is time consuming when $N$ is large. In practice, we use importance sampling to sample $p(\Theta|\boldsymbol{y}_{-i}, X, M)$ where the posterior $p(\Theta|\boldsymbol{y}, X, M)$ of the full data $\boldsymbol{y}$ is used as the proposal distribution. We thus approximate Eq. (25) as

$$\begin{aligned} p(y_i|\boldsymbol{y}_{-i}) &= \int \frac{p(y_i|\Theta)p(\Theta|\boldsymbol{y}_{-i})}{p(\Theta|\boldsymbol{y})}p(\Theta|\boldsymbol{y})d\Theta \\ &\approx \sum_{s=1}^{S} \frac{p(y_i|\Theta_s)p(\Theta_s|\boldsymbol{y}_{-i})}{p(\Theta_s|\boldsymbol{y})} \\ &\approx \frac{1}{\sum_{s=1}^{S}\frac{1}{p(y_i|\Theta_s)}} \end{aligned} \tag{26}$$

where we have omitted $X$ and $M$ in the notation for simplicity and $\Theta_s$ is a MCMC sample from the full posterior $p(\Theta|\boldsymbol{y})$. However, directly applying Eq. (26) usually results in high variance and is not recommended. We use a recently developed Pareto smoothed importance sampling to control the variance by smoothing the importance ratios $p(\Theta_s|\boldsymbol{y}_{-i})/p(\Theta_s|\boldsymbol{y})$[27,28].

The importance sampling phase is fast and it is shown to be accurate[27]. Therefore, we only need to run MCMC inference once for the full training data. Once the leave-one-out predictive probabilities in Eq. (25) are obtained for all the data points, the GP models are compared using Bayesian bootstrap described later this section.

**Stratified cross-validation**. In stratified cross-validation (SCV), we leave out all time points of an individual as test data and use the rest as training data. SCV is used when a ca/bi covariate, such as group or gender, is added to the model. Let $\boldsymbol{y}_i$ denote all measured time points corresponding to an individual $i$ ($X_i$ is defined similarly) and $\boldsymbol{y}_{-i} = \boldsymbol{y} \backslash \boldsymbol{y}_i$. Similar to LOOCV, we compute the predictive density of the test data points $\boldsymbol{y}_i$

$$p(\boldsymbol{y}_i|\boldsymbol{y}_{-i}, X, M) = \int p(\boldsymbol{y}_i|\Theta, X, M)p(\Theta|\boldsymbol{y}_{-i}, X, M)d\Theta. \tag{27}$$

This can be calculated by setting $\boldsymbol{f}^* \leftarrow \boldsymbol{y}_i$, $X^* \leftarrow X_i$, $\boldsymbol{y} \leftarrow \boldsymbol{y}_{-i}$, and $X \leftarrow X_{-i}$ in Eq. (22). Since importance sampling does not work well in this case, we apply the CCD inference $P$ times (once for each individual). Also, we use CCD with SCV as it is much faster than MCMC.

**Model comparison using Bayesian bootstrap**. After obtaining the leave-one-out predictive densities (Eqs. (25) or (27)) for a collection of models, we use Bayesian bootstrap to compare the involved models. Let us start with a simple case where two models $M_1$ and $M_2$ are compared. In the LOOCV setting, we compare the models by computing the average difference of their log-predictive densities

$$\frac{1}{N}\sum_{i=1}^{N} \left(\log(p(y_i|\boldsymbol{y}_{-i}, X, M_1)) - \log(p(y_i|\boldsymbol{y}_{-i}, X, M_2))\right), \tag{28}$$

which measures the difference of the average prediction accuracy of the two models. If Eq. (28) is greater than 0, then model $M_1$ is better than $M_2$, otherwise model $M_2$ is better than $M_1$.

Comparison in Eq. (28) does not provide a probabilistic quantification of how much better one model is compared to the other. We thus approximate the relative

probability of a model being better than another model using Bayesian bootstrap[29], which assumes $y_i$ only takes values from the observations $\mathbf{y} = (y_1, y_2, \dots y_N)^T$ and has zero probability at all other values. In Bayesian bootstrap, the probabilities of the observation values follow the $N$-dimensional Dirichlet distribution Dir(1, 1, ..., 1). More specifically, we bootstrap the samples $N_B$ times ($b = 1, \dots, N_B$) and each time we get the same $N$ observations $\mathbf{y}$, with each observation taking weight $w_{bi}$ ($i = 1, \dots, N$) from the $N$-dimensional Dirichlet distribution. The $N_B$ bootstrap samples are then summarised to obtain the probability of $M_1$ being better than $M_2$

$$\frac{1}{N_B} \sum_{b=1}^{N_B} \delta \left\{ \frac{1}{N} \sum_{i=1}^{N} w_{bi} \log \left( \frac{p(y_i | \mathbf{y}_{-i}, X, M_1)}{p(y_i | \mathbf{y}_{-i}, X, M_2)} \right) \right\}, \qquad (29)$$

where $\delta\{\cdot\}$ is the Heaviside step function and $w_{bi}$ is the bootstrap weight for the $i$th data point in the $b$th bootstrap iteration[27]. We call the result of Eq. (29) LOOCV factor (LOOCVF).

The above strategy also works when comparing multiple models. Instead of calculating the heaviside step function in the $b$th bootstrap iteration, we simply choose the model with the highest rank by sorting the models using

$$\frac{1}{N} \sum_{i=1}^{N} w_{bi} \left( \log(p(y_i | \mathbf{y}_{-i}, X, M_m)) \right), \qquad (30)$$

where $m$ indices the model. In the end, we count the occurrences $N_m$ of each model being the best across all $N_B$ bootstrap samples and we compute the posterior probability of model $M_m$ as $N_m/N_B$, which we term as the posterior rank probability.

For SCV, we replace $y_i$ with $\mathbf{y}_i$ and $\mathbf{y}_{-i}$ with $\mathbf{y}_{-i}$ in Eqs. (28, 29) and follow the same procedure as above to compare the models. Eq. (29) is then termed as the SCV factor (SCVF). In practice, we set the threshold of the LOOCVF to be 0.8 and SCVF to be 0.95, i.e. the LOOCVF (resp. SCVF) of the extended model versus the original model needs to be larger than 0.8 (resp. 0.95) for a continuous covariate (resp. bi covariate) to be added.

Although Eq. (30) can be used to compare any subset of models, complex models will dominate the posterior rank probability when compared together with simpler models. Hence, LonGP only uses it to compare candidate models of similar complexity (see next Section and Supplementary Method 2).

**Step-wise additive GP regression algorithm.** The space of all models is large and thus an exhaustive search for the best model over the whole model space would be too slow in practice. Two commonly used model (or feature) selection methods include forward and backward search techniques. Starting with the most complex model, as in the backward search approach, is not practical in our case, so we propose to use a greedy forward search approach similar to step-wise linear regression model building. That is, we start from the base model that only includes the id covariate. Then we add continuous covariates to the model sequentially until the model cannot be further improved. During each iteration, we first identify the covariate that improves the model the most (Eq. (30)) and test if the LOOCVF of a new proposed model versus the current model exceeds the threshold of 0.8 (Eq. (29)). While including a continuous covariate, we also include relevant interaction terms (allowed interaction terms defined by the user). After adding continuous covariates, we add discrete (ca or bi) covariates sequentially to the model until it cannot be further improved. As with continuous covariates, during each iteration, we first identify the discrete covariate that improves the model the most and test if the SCVF of a new proposed model versus the current model exceeds the threshold of 0.95. While including a discrete covariate, we also include relevant interaction terms (allowed interactions specified by user). Details of our forward search algorithm are given in Supplementary Method 2 together with a pseudo-algorithm description. We note that although step-wise model selection strategies are commonly used with essentially all modelling frameworks, they have the danger of overfitting a given data. To avoid overfitting, we implement our search algorithm such that an additional component is added to the current model only if the more complex model improves the model fit significantly, as measured by the LOOCVF and SCVF.

Once all the covariates have been added, the kernel parameters of the final model are sampled using MCMC and kernel-specific predictions on the training data $X$ are computed using Eq. (19). Additionally, a user can choose to exclude kernels that have a small effect size as measured by the fraction of total variance explained. We require component specific variances to be at least 1%. The software is implemented using features from the GPStuff package[24] and implementation is discussed in Supplementary Method 4.

## Data availability

All data generated or analysed during this study are included in this published article (and its supplementary information files).

## Code availability

LonGP software tool and preprocessed data sets are available at https://github.com/chengl7/LonGP.

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

## Acknowledgements

We would like to acknowledge the computational resources provided by the Aalto Science-IT and CSC-IT Center for Science, Finland. This work has been supported by the Academy of Finland Centre of Excellence in Molecular Systems Immunology and Physiology Research 2012-2017 grant 250114; the Academy of Finland grants no. 292660, 292335, 294337, 292482, 287423; JDRF grant no. 17-2013-533, 1-SRA-2017-357-Q-R; the European Union's Horizon 2020 research and innovation programme under the Marie Skłodowska-Curie grant agreement No 663830; and the Business Finland.

## Author contributions

L.C., T.V., A.V. and H.L. co-developed the method. L.C. implemented the method. S.R. performed the simulation experiments. T.V. helped with the metagenomics data set analysis. N.L. and R.L. helped with the proteomics data set analysis. H.L. oversaw the whole process. All authors contributed to the manuscript writing.

## Additional information

**Competing interests:** The authors declare no competing interests.

