## [Peer Review File · Nature Communications]

Reviewers' comments:

Reviewer #1 (Remarks to the Author):

The paper describes a new additive Gaussian process (GP) model for longitudinal data analysis. Hypotheses are encoded as alternative models which can be used to test health-related effects but control for other major sources of variations. By using a GP framework the authors can include non-linear effects and temporally-correlated "noise" - i.e. individual-specific effects. The inclusion of non-linear individual-specific effects is a powerful aspect of such additive GP models. State-of-the-art Bayesian model selection procedures are used to select the best model (using an LOOCV procedure) and therefore the best scientific hypothesis for sources of variation in each patient or measurement. Inference is carried out using modern Bayesian procedures allowing the method to be applied to practical examples.

Careful simulations are carried out which demonstrate improved performance over standard mixed effects models and an ARD GP (which uses a less rigorous approach to covariance model selection).

Two applications are used to demonstrate the methods usefulness. In the first application results are similar to a previous approach but in the Type-1 diabetes application more proteins related to the condition are identified.

Overall I found this to be a high-quality and useful contribution to the statistical modelling toolbox. Methods were state-of-the-art and the simulations and applications show the potential to extract improved hypotheses from longitudinal datasets.

Corrections

In the proteomics dataset it was not clear how different the results are from the previous analysis. Some description of how different the proteins are which were discovered, along with evidence that the new discoveries are scientifically plausible, would be useful in this example.

Minor typos and style issues

P3: After eqn (1) "also called kernel" could be "also called the kernel function"

P3: "and Methods Section" - "and the Methods Section"

P8: "adjusted to a given data" - "adjusted to the given data"

Reviewer #2 (Remarks to the Author):

This is an interesting and well-written paper on a new statistical approach called LonGP which seems to allow for a more flexible and robust modelling of longitudinal data compared to the commonly used GLM modelling. The authors did well to apply LonGP to both a simulated dataset and two independent real-life datasets. As an applied user, I can imagine the potential of this new approach.

However, I have two serious concerns regarding applicability:

1. The fact that the LonGP software can only be used in Matlab and particularly only on Linux is quite a severe restriction. Would it be possible to implement LonGP into an R package which runs

on all major operating systems?

2. I wonder whether LonGP also allows to quantify uncertainty of the estimates, e.g. by adding confidence bands to the plots.

Additional minor issues are listed below:

Introduction:

- It may be worth to extend the paragraph which introduces GP modelling in a way that it describes the idea / concept of GP to applied researchers who are not familiar with GP. The authors might also explain there why this is considered to be a non-parametric approach, as the word "Gaussian" seems to imply that it somehow relies to a normal distribution assumption.
- "LonGP also incorporates non-stationary signals...": Please explain what this means and why this is an advantage compared to GLMs.

Results:

- Tables / Figures: As there are so many abbreviations used throughout the manuscript, I suggest to explain them again in each table/figure legend. Additionally, each legend should mention to which dataset the results belong, and the axis labels in most figures need to be increased substantially.
- What is meant by a "sero" effect / component? This is somehow explained in 2.4., but I don't understand its meaning in the context of the other two examples.
- Figure 2: Suggest to extend the y-axis to 0-100% in order to make the figure better comparable to figure 3.
- The GP-ARD approach seems to come out of the blue. It would be good to give some context for this and a rationale for taking it into consideration.
- "We require a pathway to be detected in at least 500 samples to be included... which results in 394 microbial pathways". I do not understand this sentence. Does this follow from a specific condition of the data, from previous analyses, or is this an anticipation of the results shown later?
- I assume the "id" variable identifies each individual? Suggest to mention this.
- How was seroconversion in the T1D dataset defined: As first presence of single or multiple autoantibodies? Which autoantibodies were taken into account?
- How do the results achieved with LonGP differ from those in Liu et al.? Do the LonGP results make sense from a medical/biological perspective?

Discussion:

- It would be good to have a statement why exactly LonGP seems to be superior to LME methods in modelling the simulated data. What exactly was the condition in the simulated data which makes them difficult (or even inappropriate?) for LME modelling?
- Would it be possible to add a routine to the LonGP software tool which allows sample size calculation based on certain assumptions?

Supplementary Information:

- Which data will exactly be made available online?

Andreas Beyerlein, PhD
Institute of Computational Biology
Helmholtz Zentrum München, Neuherberg, Germany

Reviewers' comments:

Reviewer #1 (Remarks to the Author):

The paper describes a new additive Gaussian process (GP) model for longitudinal data analysis. Hypotheses are encoded as alternative models which can be used to test health-related effects but control for other major sources of variations. By using a GP framework the authors can include non-linear effects and temporally-correlated "noise" - i.e. individual-specific effects. The inclusion of non-linear individual-specific effects is a powerful aspect of such additive GP models. State-of-the-art Bayesian model selection procedures are used to select the best model (using an LOOCV procedure) and therefore the best scientific hypothesis for sources of variation in each patient or measurement. Inference is carried out using modern Bayesian procedures allowing the method to be applied to practical examples.

Careful simulations are carried out which demonstrate improved performance over standard mixed effects models and an ARD GP (which uses a less rigorous approach to covariance model selection).

Two applications are used to demonstrate the methods usefulness. In the first application results are similar to a previous approach but in the Type-1 diabetes application more proteins related to the condition are identified.

Overall I found this to be a high-quality and useful contribution to the statistical modelling toolbox. Methods were state-of-the-art and the simulations and applications show the potential to extract improved hypotheses from longitudinal datasets.

Corrections

In the proteomics dataset it was not clear how different the results are from the previous analysis. Some description of how different the proteins are which were discovered, along with evidence that the new discoveries are scientifically plausible, would be useful in this example.

Response: We have now included a comparison of proteins identified in our analysis with the results reported in (Liu et al, 2018) (added to page 8, Section 2.4) and discuss the differences between the results in Sections 2.4 and 3. Additionally, we provide biological interpretations and relevance for some of the newly discovered proteins, in particular secretogranin-3 (Q8WXD2) and FAM3C (Q92520) in page 9, Section 3.

Minor typos and style issues

P3: After eqn (1) "also called kernel" could be "also called the kernel function"

P3: "and Methods Section" - "and the Methods Section"

P8: "adjusted to a given data" - "adjusted to the given data"

Response: Corrected.

Reviewer #2 (Remarks to the Author):

This is an interesting and well-written paper on a new statistical approach called LonGP which seems to allow for a more flexible and robust modelling of longitudinal data compared to the commonly used GLM modelling. The authors did well to apply LonGP to both a simulated dataset and two independent real-life datasets. As an applied user, I can imagine the potential of this new approach.

However, I have two serious concerns regarding applicability:

1. The fact that the LonGP software can only be used in Matlab and particularly only on Linux is quite a severe restriction. Would it be possible to implement LonGP into an R package which runs on all major operating systems?

Response: LonGP is dependent on another Matlab package called GPstuff. GPstuff only supports Matlab (also Octave, a free variant of Matlab) in Linux and macOS systems. For convenience of the users, we have provided two additional solutions

(1) Compiled versions of LonGP to be run in Linux and macOS operating systems.

The user only needs to install Matlab MCR (freely available from Mathworks) to run LonGP. Although the installation process is simple, we provide instructions:

<https://github.com/chengl7/LonGP#install-compiled-longp>

(2) Octave version of LonGP. Octave is an open source (free) version of Matlab. We have modified the LonGP code such that it also supports Octave. Details:

https://github.com/chengl7/LonGP/tree/LonGP_Octave

So, our LonGP method can now be used in both Linux and macOS operating systems using Matlab, Octave or compiled code.

2. I wonder whether LonGP also allows to quantify uncertainty of the estimates, e.g. by adding confidence bands to the plots.

Response: Yes, as the nature of GP, we can obtain the predictive distribution at each given data point. These uncertainty estimates also incorporate uncertainty in the kernel parameters as shown in Eqs. (19) and (22). We have added a note on that in Section 3. We have also added two examples in Supplementary Materials, where we provide the version with +/- 1 standard deviation around the mean of Fig. 4b) and 4c) (Figure 8 and 9 in the supplementary). In addition, we have updated the documentation in github so that a user can incorporate the uncertainty estimates in his/her figures too.

Additional minor issues are listed below:

Introduction:

- It may be worth to extend the paragraph which introduces GP modelling in a way that it describes the idea / concept of GP to applied researchers who are not familiar with GP. The authors might also explain there why this is considered to be a non-parametric approach, as the word "Gaussian" seems to imply that it somehow relies to a normal distribution assumption.

Response: Following reviewer's suggestion, we have revised and extended the general (non-technical) introduction to GP modeling in Section 2.1. We also better connect the non-technical description to an illustration shown in Figure 1. We also briefly discuss the non-parametric nature of GPs. As we cannot cover this latter aspect in full in the manuscript, we cite relevant material for further reading.

- "LonGP also incorporates non-stationary signals...": Please explain what this means and why this is an advantage compared to GLMs.

Response: This particular sentence was not clearly written, so we changed it to: "LonGP also models non-stationary signals using non-stationary kernel functions and...". The non-stationary refers to the non-stationary kernel in Section 4.4.2, which allows us to model longitudinal phenomenon whose statistical properties are not time-shift invariant. For example, pathophysiological mechanisms/changes can have faster dynamics around a disease onset time than changes at other time points. Non-stationary kernel functions provide a way to model such signals, and we have implemented them into our LonGP model. While it may in principle be possible to model non-stationary signals with linear models that would be challenging as one would need to choose an explicitly parameterized non-stationary effects in a form appropriate for linear models. Whereas in GP formulation, non-stationary regression with Bayesian inference can be conveniently formulated and implemented using non-stationary kernel functions, as we have done in our manuscript. We have added a discussion of these points into Discussion Section.

Results:

- Tables / Figures: As there are so many abbreviations used throughout the manuscript, I suggest to explain them again in each table/figure legend. Additionally, each legend should mention to which dataset the results belong, and the axis labels in most figures need to be increased substantially.

Response: Corrected as suggested: we have increased the font sizes in Figures 1, 4, and 5, and we added the dataset information and explained abbreviations in each figure/table legend.

- What is meant by a "sero" effect / component? This is somehow explained in 2.4., but I don't understand its meaning in the context of the other two examples.

Response: First, "sero age" of an individual is defined as the time relative to the age of the first detection of T1D autoantibodies. Sero effect / component then refers to a Gaussian process effect / component that is a function of sero age and is consistent across all cases. An illustrative sero effect is shown in Figure 1 (top panel, third figure). Note that the age at which the T1D autoantibodies are detected is different for each individual. For each individual, the Gaussian process sero effect is then "localized" at the individual-specific seroconversion time point, making the sero effect consistent in the "sero age" coordinate but difficult / impossible to detect in the absolute age coordinate. The sero effect aims to detect non-linear and non-stationary effects that appear at specific times before / after the seroconversion, possibly near the time of the seroconversion. As with any additive GP components in our model, the sero effect has no parametric forms and is modelled by the non-stationary kernel described in Section 4.4.2. We have incorporated parts of the above description in Section 2.4.

- Figure 2: Suggest to extend the y-axis to 0-100% in order to make the figure better comparable to figure 3.

Response: Updated according to suggestion.

- The GP-ARD approach seems to come out of the blue. It would be good to give some context for this and a rationale for taking it into consideration.

Response: We chose to include GP-ARD in performance comparisons because it is the most commonly used method for assessing relevance of variables in GP regression. We have revised our manuscript by giving background and motivation for the use of GP-ARD (Section 2.2).

- "We require a pathway to be detected in at least 500 samples to be included... which results in 394 microbial pathways". I do not understand this sentence. Does this follow from a specific condition of the data, from previous analyses, or is this an anticipation of the results shown later?

Response: This is just a prefiltering step that allows use to focus our analysis on those microbial pathways that are sufficiently abundant, or in other words, do not have excessive amount of "missing data" (i.e., samples with no sequencing reads mapped to genes in a given pathway). To clarify this we have revised the sentence in Section 2.3 as

follows: "To focus our analysis on pathways with sufficiently strong signal, we include in our analysis pathways that have been detected (i.e., at least one sequence read maps to genes of a pathway) in at least ~64% (=500/785) of the samples."

- I assume the "id" variable identifies each individual? Suggest to mention this.

Response: Correct. We have clarified that in Section 2.1.

- How was seroconversion in the T1D dataset defined: As first presence of single or multiple autoantibodies? Which autoantibodies were taken into account?

Response: In our analyses we have used the seroconversion times reported by Liu et al. (2018) and have now added a reference to that publication in Section 2.4 where the concept of seroconversion is introduced. According to Liu et al. islet autoimmunity is defined as the presence of one or more of the autoantibodies (here IAA, GADA, IA-2A and ZnT8A measured) on at least two consecutive visits 3-6 months apart or development of T1D within 6 months after a positive autoantibody test (Liu et al. 2018, p. 101). In their study the time point of seroconversion to multiple islet autoantibodies was selected to be in the middle of the time series for T1D patients (Liu et al. p. 103), and the seroconversion times for each T1D patient are shown in Figure 2 and in Supplementary Table S2 of the article.

- How do the results achieved with LonGP differ from those in Liu et al.? Do the LonGP results make sense from a medical/biological perspective?

Response: We have added comments on the differences between the original analyses by Liu et al. and LonGP analyses in Sections 2.4. and 3. We have also added discussion about the biological relevance of some of the novel findings in Section 3.

Discussion:

- It would be good to have a statement why exactly LonGP seems to be superior to LME methods in modelling the simulated data. What exactly was the condition in the simulated data which makes them difficult (or even inappropriate?) for LME modelling?

Response: LonGP has at least three features which makes it more efficient in our simulated data scenarios than the standard LME model. First, kernels automatically implement arbitrary non-linear effects, whereas LME model is limited to linear (or second-order polynomial) effects. This is accentuated by having several non-linear effects for individual covariates or their interactions. Moreover, characterizing the posterior of the kernel parameters further improves LonGP's ability to identify non-linear

effects: instead of optimizing the kernel parameters to a given data set we also infer their uncertainty, and thus improve predicting new/unseen data points and inferring the covariate effects at the end. Second, LonGP contains non-stationary effects that can be difficult to model using linear models. Third, LonGP naturally implements individual-specific time-varying random effects, which we consider relevant in modeling real biomedical longitudinal data sets, too. We have included the above comments into Discussion Section.

- Would it be possible to add a routine to the LonGP software tool which allows sample size calculation based on certain assumptions?

Response: We agree that sample size calculations for additive Gaussian processes would be an useful addition for our manuscript but that is beyond the scope of this manuscript and we leave it as part of future work. Our results using simulated data give some understanding of how sample size affects the inference results for varying amounts of noise and varying sample sizes.

Supplementary Information:

- Which data will exactly be made available online?

Response: We have uploaded to the LonGP github page the preprocessed files of the metagenomics dataset and proteomics dataset used in this study:

<https://github.com/chengl7/LonGP/tree/master/datasets>

The LonGP results of the metagenomic dataset and proteomics dataset are included in the supplementary file.

Andreas Beyerlein, PhD
Institute of Computational Biology
Helmholtz Zentrum München, Neuherberg, Germany

REVIEWERS' COMMENTS:

Reviewer #2 (Remarks to the Author):

The authors have responded convincingly to my issues. I would still like to see getting LonGP implemented into R, but this can be considered to be future work. The same applies to a sample size calculation routine which would be useful in my eyes, but I understand that this is out of the scope of this paper.